# Diagnostic Evaluation and Preparation of the Reference Information for River Restoration in South Korea

**DOI:** 10.3390/ijerph18041724

**Published:** 2021-02-10

**Authors:** Chi Hong Lim, Jeong Hoon Pi, A Reum Kim, Hyun Je Cho, Kyu Song Lee, Young Han You, Kye Han Lee, Kee Dae Kim, Jeong Suk Moon, Chang Seok Lee

**Affiliations:** 1National Institute of Ecology, Seocheon 33657, Korea; sync03@nie.re.kr; 2Korea National Baekdudaegan Aboretum, Bonghwa 36209, Korea; ecopjh@korea.kr; 3Department of Bio & Environmental Technology, Seoul Women’s University, Seoul 01797, Korea; dkfma@swu.ac.kr; 4School of Forest Science & Landscape Architecture, Kyungpook National University, Daegu 41566, Korea; jhj132@hanmail.net; 5Department of Biology, Gangnung-Wonju University, Gangnung 25457, Korea; leeks84@gwnu.ac.kr; 6Department of Biology, Kongju National University, Kongju 32588, Korea; youeco21@kongju.ac.kr; 7Department of Forest Resources, Chonnam University, Gwangju 61186, Korea; khL@jnu.ac.kr; 8Department of Environmental Education, Korea National University of Education, Cheongju 28173, Korea; kdkim@knue.ac.kr; 9National Institute of Environmental Research, Incheon 22689, Korea; waterfa@korea.kr

**Keywords:** diagnostic evaluation, reference river information, restoration, riparian vegetation, river type

## Abstract

We assessed the naturalness of rivers based on the riparian vegetation index throughout the national territory of South Korea as a preparatory process for restoration to improve the ecological quality of rivers. The riparian vegetation index was obtained by incorporating the diversity of species and community, vegetation profile, and ratios of the number of species and areas occupied by exotic, obligate upland, and annual plants. The evaluation was conducted based on both the riparian vegetation index and each vegetation component. The result of the evaluation based on the riparian vegetation index showed that more than 70% of the river reaches were graded as less than “moderate” and exotic and obligate upland plants were more common than endemic aquatic plants. The reaches recorded as “very good” and “good” grades were usually restricted around the upstream of the north and central-eastern parts, whereas reaches of the other areas showed “poor” naturalness (less than “moderate”). The vegetation components selected for the evaluation showed a significant correlation with each other as well as the riparian vegetation index. The degree of contribution of each vegetation component showed that the vegetation profile played the most significant role, followed by species diversity, community diversity, and the ratio of area occupied by annual plants. The riparian vegetation index revealed a significant correlation with the indices based on other taxa such as benthic invertebrates, periphytic algae, and fish, habitat conditions in the waterway, and water quality based on biochemical oxygen demand (BOD). The diagnostic evaluation results imply that most reaches need ecological restoration. The reference information was prepared by incorporating the vegetation condition with the highest score in each reach in the diagnostic evaluation. The river reach was divided into five reaches of upper and lower valley streams, upstream, midstream, and downstream. Information on the reference vegetation for restoration was prepared with the stand profile including both horizontal and vertical arrangements of riparian vegetation and species composition classified by the reach divided into five types. The levels of restoration were determined based on the diagnostic evaluation results. The lower the naturalness grade, the higher the level of restorative treatment was recommended.

## 1. Introduction

Rivers are a continent’s circulatory system and the study of rivers, much like the study of blood, can be used to diagnose the health of not only the rivers themselves but also their landscapes [1,2]. A biota that evolves and maintains itself for a long time in a region possesses biological integrity—the capacity to support and maintain a balanced, integrated, and adaptive biological system with a full range of elements and processes that are expected in a region’s natural habitat [3,4,5,6,7].

Systems with biological integrity can withstand or recover rapidly from most natural disturbances [8,9,10]. However, the biological integrity declines if a natural disturbance regime is altered in intensity, type, or frequency—especially if the disturbances become incessant. Urbanization, for example, compromises the biological integrity of rivers by severing the connections in a watershed and altering hydrology, water quality, energy sources, habitat structure, and biotic interactions [9,11,12]. 

Riparian ecosystems are spatially and temporally dynamic and thereby shaped by fluvial geomorphic processes. Therefore, physical and biological links exist between the terrestrial and aquatic environments and biotopes in which animals may seek refuge and food, while enriching the soil with detritus [13,14,15]. A riparian habitat supports the surrounding fluvial ecosystem throughout its entire length and integrates many interactions between the aquatic and terrestrial components of the landscape. Therefore, riparian habitats are crucial for the preservation of river biodiversity [16,17,18,19]. In fact, riparian ecosystems usually support higher species richness and wildlife density than other nearby ecosystems [20,21,22,23]. Riparian systems also represent a vital component of river management because their state affects many ecological services related to the river. Because a riparian zone is spatially close and connected to the waterway, riparian systems are flooded periodically and play an important role in water infiltration and aquifer recharge. Moreover, riparian vegetation provides flood attenuation and serves to decrease hydrological risks, detain erosion materials, and reinforce stream banks with root systems; thus, reducing the amount of solids suspended in watercourses and improving water quality [24,25,26,27,28,29,30,31,32]. In addition, riparian vegetation, as an important landform agent and flow resistance factor, is responsible for most of the energy losses in fluvial systems. Roots increase substrate cohesion, and stems and leaves modify riverbed roughness, thereby controlling erosion and sediment transport and deposition (both in the channel and floodplain) [33,34,35,36]. Several processes for the exchange of matter and energy in the river channel occur in the riparian zone. This habitat serves to protect in-stream water quality by acting as a sink and filter of sediment and nutrients [18,37,38]. Moreover, riparian forests represent important natural corridors in the landscape and constitute areas of high biodiversity. These forested corridors have great value as sites for recreational and cultural events [39]. 

These advantages, together with the considerable enhancement of the landscape, make riparian vegetation of primary importance. Therefore, maintenance and/or restoration of riparian vegetation needs to be prioritized in land-management projects [18,30,40,41].

The range and status of riparian vegetation is not only related to the integrity index of rivers evaluated through the macroinvertebrate index, which was commonly used, but also affects the overall of the riverine landscape attributes. In this respect, the management of the riparian vegetation is of paramount importance to conserve biodiversity in river ecosystems at a time of rapid environmental change and increasing pressure from a range of human influences, including habitat loss, pollution, and climate change [42].

In order to understand the condition of riparian systems and what, if any factors, promote or degrade condition, specific tools are required. Having tools that can spatially locate good and poor condition reaches can assist with the prioritisation of sections of rivers for improvement in condition and aid in the development of specific management strategies [43]. Various methods have been proposed to evaluate the riparian conditions of rivers. Vegetation structure [44,45,46], riparian dimensions, habitat quality, and land use [15,47,48] have been usually used for assessing riparian quality. Other river assessment methods also use some riparian characteristics to assess the status of the physical habitat, depending on the objectives [49,50]. Various indices developed by incorporating such environmental factors have been proposed to evaluate the riparian conditions of rivers. Riparian health indicators and pressure indicators, vegetation state, and morphological condition have been usually adopted [15,51,52]. Qualitat del Bosc de Ribera (QBR) Index considers key aspects of vegetation, such as coverage and structure, as well as morphological aspects and anthropogenic interference in the landscape [53]. The Riparian Strip Quality Index (RSQI) evaluates the ecological condition of riparian habitats based on the percentage of riparian vegetation [42,52]. The Riparian Condition Index (RCI) considers five components such as riparian vegetation cover, biodiversity condition, hydrologic stress, morphological condition, and catchment disturbance. The Riparian Condition Index (RCI) evaluates the aspects of the riparian zone that might be indicative of poor quality and integrity. Attributes are graded from poor (1) to excellent (5) on the assessment reach, and scores were summed to get the index [43]. The Riparian Quality Index (RQI) evaluates the structure of riparian zones and the river morphological conditions. Width of riparian corridor, longitudinal continuity, coverage and distribution pattern of riparian corridor, composition and structure of riparian vegetation, age diversity and natural regeneration of woody species, bank conditions, floods and lateral connectivity, and substratum and vertical connectivity are considered for obtaining the index. Based on the index value, the riparian status and the river management options are determined [15].

The riverine ecosystems are increasingly under threat as they are confronted with various anthropogenic interferences. Comprehensive management strategies are required to cope with, or prevent, long-term impacts on habitats and their biodiversity, as well as on their ecological functions and services. The basis for the efficient management and effective conservation of any ecosystem is sufficient information on the state of the system and its response to influential external factors. In a riverine ecosystem, state information is currently drawn from ecological assessments at the reach or site scale. While these assessments are essential, they are not sufficient to assess the expected outcome of different river management strategies, because they do not account for important characteristics of the whole riverine ecosystems. Efficient and effective management of riverine environments could be supported best by integrative national-scale ecological assessments. This is of particular importance for the spatial prioritization of management measures. Assessments at this scale are of increasing interest to environmental managers and conservation practitioners to prioritize management measures or to locate areas worth protecting [54]. 

We present an approach that integrates ecological assessments on both health and pressure of riparian vegetation to describe spatially the ecological state of all the rivers throughout the entire national territory of South Korea. This approach adopted bio-, eco-, and functional diversities as the riparian health indicator and considered percentage based on the number of species and occupied area of the exotic, obligate upland, and annual plants as the stress factor. 

In Korea, where rice is used as a staple food, most of floodplains were transformed to rice fields, and big banks were constructed to prevent flood damage. As a result, the widths of most rivers and/or streams were reduced greatly. Furthermore, they were constantly exposed to and damaged by human interference in the process of supplying water to the rice paddies, and in modern times, they are controlled by more human interference due to urbanization. Consequently, meandering and complex channels were changed to straight and monotonous ones [55,56]. In reflection of this reality, our assessment focused on the riparian vegetation rather than on the morphology of the river in this study.

Biodiversity, which encompasses genes, species, ecosystems, and their interactions, can be an indicator of integrated and healthy ecosystems because greater biodiversity ensures sustainability and stability [57,58,59]. Because biodiversity is always tied to habitats, habitat diversity or eco-diversity becomes the basis for establishing biodiversity [60]. 

Considered these aspects, we performed a diagnostic evaluation of rivers with regard to the diversities of species, community, and stand profile, which reflect habitat diversity. In addition, we selected exotic, obligate upland, and annual plants as variables for the diagnostic assessment to reflect the disturbance effects.

The assessment of the river was begun with a simple survey centered on flood defense. These surveys were very important in terms of river engineering, but they were not sufficient to objectively and extensively assess the physical state of the river. In 1992, the European Commission developed a radically new approach to river management as an early precursor of the European Water Framework Directive (hereafter abbreviated as WFD). It was a fully ecological approach rather than just chemical and biological assessment of rivers and it was a way of characterising the physical structure of rivers and assessing how this affected biological communities [61]. The WFD aims at enhancing the status of aquatic ecosystems including rivers and biotic communities in a comprehensive way. Water management is brought beyond water quantity and quality, entailing provisions on land-use and governance. The WFD sets environmental objectives in terms of good status to be met by 2015, or under certain conditions the final deadline of 2027 [62].

The river evaluation is also an important first step in the restoration process. The obtained data can be used to determine the need and potential for restoration and are essential for the development of a plan. The data collected through a quantitative and qualitative investigation of a river and riparian ecosystem are used to determine if the stream is evolving towards stability or instability and if the cause of instability is localized or system-wide [63,64].

In this study, we aimed to diagnose all the rivers in the national territory of South Korea based on riparian vegetation data as a preparatory process for river restoration. Furthermore, we prepared the reference information for restoration of the degraded river by incorporating data on riparian vegetation collected from the reaches, which maintain relatively integrated riparian vegetation, evaluated as “very good grade” in this study. Finally, we suggested a restoration plan based on the results of the evaluation.

## 2. Materials and Methods 

### 2.1. Study Area

This study was performed in all river reaches divided at regular intervals throughout the national territory of South Korea (refer to Figure 1). 

### 2.2. Data Sampling

A vegetation survey to evaluate the integrity of rivers was conducted from May to October 2013 and 2014 at 960 locations across the entire national territory of South Korea. We constructed a vegetation map over a distance of 200 to 1000 m, depending on the river breadth. Aerial photo images were used to identify the vegetation types and landscape boundaries. These vegetation types and landscape elements were confirmed by field checks. Landscape attributes were overlapped onto topographical maps at 1:5000 scales. Patches smaller than 1 mm on the map were excluded from this study because of the uncertainty of their sizes and shapes [65]. Mapping was performed using the ArcView GIS (Geographic Information System), and landscape ecological analyses were conducted with the ArcView GIS software [66].

A vegetation profile was prepared by carefully depicting the micro-topography and major plant species in a belt transect installed in 10-m widths between levees on both sides of the rivers. 

Occurrence and dominance were recorded for all plant species appeared in field plots installed randomly on the spatial range that the vegetation map was constructed [67,68]. Plot sizes were 1 × 1 m in the riparian plains dominated by herbaceous vegetation immediately adjacent to stream channels, 5 × 5 m in the shrub lands, and 10 × 10 m in the forests distant from the stream channels. Nomenclature followed Lee [69] and Korea National Arboretum [70]. Dominance was estimated with the Braun–Blanquet [67] ordinal scale from 1 (<1%) to 5 (>75%). 

### 2.3. Data Processing

The naturalness in rivers was evaluated from the perspectives of both each vegetation component including species diversity, community diversity, vegetation profile, and the number and occupied area ratios of exotic, obligate upland, and annual plant species, and the riparian vegetation index obtained by incorporating the results evaluated on each component. The community diversity and occupied areas of exotic, obligate upland, and annual plants were obtained from vegetation maps. The community diversity was based on the number of plant communities expressed on the vegetation map. The occupied area ratios for exotic, annual, and obligate upland plants were obtained from the percentage of the area that they dominated to the total area. The number of species ratios for exotic, annual, and obligate upland plants were obtained from the percentage of the number of species to the total number of species. Naturalness based on the vegetation profile was evaluated in terms of the assessment of functional diversity. Naturalness based on the vegetation profile was evaluated based on the response of the vegetation to natural and artificial disturbances, according to Lee and You [71]. The species diversity was based on the number of species surveyed. The score for each vegetation component, ranging from 1 (lowest) to 5 (highest) was provided by dividing, at the same interval, the range between the highest and lowest values of each component collected at regular intervals throughout the country (Table 1). 

The weighted values of each vegetation component were determined with the aid of experts who participated in a national project to evaluate the integrity of the rivers’ ecosystems. A weighted value of 1 point was given to the percentage based on the number of species and occupied areas of exotic, annual, and obligate upland plants. We assigned weighted values of 2 and 4 points for species diversity, which addresses the composite factor related to various species, and community diversity, which addresses the composite factor related to various vegetation types as a two-dimensional component, respectively. The vegetation profile expresses the horizontal and vertical diversity of vegetation; 8 points were conferred to this component. The riparian vegetation index was obtained from the sum of the scores multiplied by the weighted value of each vegetation component. The riparian vegetation index was divided into five grades of “very good,” “good,” “moderate,” “poor,” and “very poor” (Table 2). 

We obtained information on the naturalness in rivers based on data on benthic invertebrates, fish, periphytic algae, habitat conditions in the waterway, and biochemical oxygen demand (BOD) from the Ministry of Environment [72].

Differences in the naturalness grade among the reaches were analyzed with nonmetric multidimensional scaling (NMDS) stand ordination [73], which was performed using PC-Ord 4.0 [74]. The vegetation components with weighted value for each reach were fed into a matrix for the NMDS ordination. 

### 2.4. Preparation of the Reference Information for Restoration of the Degraded River

The reference condition was prepared by classifying the river into five reaches of downstream, midstream, upstream, lower valley stream, and upper valley stream based on the slope of the riverbed, as it dominates the flow speed of water and further determines riverbed substances and riparian vegetation type. We assumed the reference condition as that of the reach with the highest vegetation index score. The level of ecological restoration required was determined by comparing the riparian vegetation index of each river reach with the reference condition of the reach. The horizontal range was restricted within the riverbank in this restoration plan because most reaches are blocked by the riverbank for protection against flooding in the transformed land around the river.

### 2.5. Statistical Analysis

Relationships between the riparian vegetation index and each vegetation component, indices based on other taxa including benthic invertebrates, fish, periphytic algae, habitat conditions in the waterway, and BOD were quantified using Spearman’s correlation coefficient. The statistical analyses were performed using SPSS 19 [75].

## 3. Results

### 3.1. Diagnostic Evaluation of Rivers 

The spatial distribution of the grade based on each vegetation component and the riparian vegetation index were expressed as maps (Figure 1). The results of diagnostic evaluation based on the number of exotic, obligate upland, and annual plant species, vegetation profile, and species and vegetation diversities usually showed a low grade, whereas those on the areas occupied by exotic, annual, and obligate upland plants revealed a relatively high grade (Figure 1). The riparian vegetation index—sum of values obtained from multiplying the score and weighted value of each vegetation component—showed a relatively poor grade (Figure 1). 

In a map based on species diversity, reaches evaluated as “very good” and “good” grades were usually distributed in the northern and eastern parts, whereas reaches evaluated as “very poor” and “poor” grades were in the southern and western parts. The maps based on the number of exotic plant species and vegetation profile usually showed poor grades. In a map based on the number of annual plant species, reaches evaluated as “very good” and “good” grades were usually distributed in the northeastern and southern parts, those evaluated as “very poor” and “poor” grades were in the western and central eastern parts. In a map based on the number of obligate upland plant species, reaches evaluated as “very good” and “good” grades were usually distributed in the central and northwestern parts, whereas those as evaluated “very poor” and “poor” grades were in the other parts. The maps that expressed the results evaluated on the areas occupied by exotic and annual plants usually had high grades, except in some reaches. In a map that expressed the results evaluated on the area occupied by obligate upland plants, reaches evaluated as “very good” and “good” grades were located in upstream originated from the Baekdudaegan, which is the representative mountains in Korea, stretched in the north to south but the south-western directions in the northern and in the southern parts. Meanwhile, midstream and downstream reaches centering on the western part showed relatively poor grades. The map based on community diversity showed that reaches evaluated as “very good” and “good” grades were usually distributed in the central and southwestern parts, whereas reaches located in the other parts showed relatively low grades. In a map based on the riparian vegetation index, reaches evaluated as “very good” were very rare and those evaluated as “good” grade were usually upstream reaches in the north and central eastern parts. On the other hand, reaches in the other parts had poor grades less than “moderate.”

### 3.2. Relationships between the Riparian Vegetation Index and Grades Based on Each Vegetation Component 

The vegetation components used for the diagnostic evaluation of the rivers showed significant correlations with each other and the riparian vegetation index (Table 3). Among the correlations between each vegetation component, the relationships between the areas occupied by the obligate upland and exotic plants were the closest, followed by the relationships between the number of species and occupied area of annual plants, species diversity and community diversity, and occupied areas of annual and obligate upland plants. 

The riparian vegetation index showed the closest relationship with the vegetation profile, followed by species diversity, community diversity, and area occupied by annual plants.

### 3.3. Relationships between the Riparian Vegetation Index and Indices Based on Other Taxa and Water Quality

The riparian vegetation index had significant correlations with the indices based on other taxa such as benthic invertebrates, periphytic algae, fish, habitat conditions of the waterway, and water quality based on BOD (Table 4). This result reflects the suitability of diagnosing rivers based on the riparian vegetation index.

As the result of stand ordination based on score of each vegetation component, reaches evaluated as “very good” or “good” reaches tended to be arranged in the lower part, whereas the “very poor” or “poor” reaches were located in the upper part of Axis 2 (Figure 2). Community diversity, vegetation profile, occupied area of exotic plants, and occupied area of obligate upland plants dominated the arrangement of stands; the former two factors affected the distribution of “very good” or “good” reaches, whereas the latter two factors affected the distribution of “very poor” or “poor” reaches.

### 3.4. Reference Information for Restoration of the Degraded River

Information on the reference vegetation was prepared with the stand profile including both horizontal and vertical arrangements of riparian vegetation (Figure 3) and species composition classified by the reach (Table 5). In the valley-stream reaches, the period of submersion at flooding is relatively short because of the steep slope of the riverbed and coarse substrate of the bed. Therefore, there are easily drained, and upland terrestrial plants dominate the riparian zone of the reaches. The breadth of the streams induces the difference between the upper and lower valley streams. Lower valley streams with relatively broader breadth retain the shrub-dominated zone, which is composed of shade intolerant species and is different from upper valley streams without the zone. Upstream and midstream maintain the integrated arrangement of riparian components including bare ground, herb, shrub, and tree-dominated zones. However, the species compositions of both reaches are distinct because of differences in current speed and substrate particle size. For example, the ratio of plants that favor substrates with coarse particles, including upland plants, is higher in upstream than in midstream. However, the boundary between the shrub- and tree-dominated zones is obscure in downstream because the slow current speed causes the differences in the disturbance regime between the zones to disappear.

### 3.5. Restoration Plan Based on the Diagnostic Evaluation Results

The levels of restoration determined based on the diagnostic evaluation results are listed in Table 6, and the restoration plans based on the grade are shown as diagrams in Figure 4. The lower the naturalness grade, the higher is the level of recommended restorative treatment. The restoration methods were based on various studies [41,76,77,78] related to ecological restoration, including “The SER International primer on ecological restoration” [63,64]. 

Usually, the river reaches evaluated as “very poor,” are channelized rather than meandering longitudinally and equipped with a terraced structure rather than a puddle type with a gentle horizontal gradient. Moreover, a relatively wide range is covered with various artificial spaces. Therefore, exotic, obligate upland, and annual plants dominate the vegetation established there. Consequently, both community and species diversities are very low, and the vegetation profile is very simple. Because of the poor ecological conditions, the method of restoration should be active. The range of restorative treatments has to cover the entire range of the riparian landscape from the channel through the floodplain to the weir, and further should take into account river morphology, vegetation, watershed management, and networking with the surrounding terrestrial ecosystem (Table 6). First, the terraced structure should be transformed into a riverine structure with the waterway, floodplain, and weir naturally connected with each other by a gentle gradient that imitates a natural river. Second, the rivers should be connected to the surrounding terrestrial ecosystems through the ecological network. With regard to water quality, we have to control land use in the watershed. Furthermore, the channelized waterway should be transformed into a meandering one. Introduction of vegetation should be performed actively. Herb-, shrub-, and tree-dominated vegetation should be established in the mentioned order along the distance from the waterway through the floodplain to the weir. The vegetation should be introduced by using a natural river as reference (Figure 4). 

The ecological condition of river reaches evaluated as “poor” was similar to that of river reaches evaluated as “very poor.” However, the vegetation coverage increased with the establishment of perennial plants; therefore, species as well as community diversities increased, and the vegetation profile improved a little. Therefore, restorative treatment similar to that in the river reaches evaluated as “very poor” is required, but the treatment level can be lower (Table 6). 

The ecological condition of river reaches evaluated as “moderate” was relatively improved, as woody as well as perennial plants were present. This decreases the ratio of areas occupied by exotic and obligate upland plants. Consequently, both community and species diversities increased, and the vegetation profile improved. However, improvement of the riverine structure from a non-ecological terraced frame to an ecological puddle type is still required, and reinforcement of vegetation is necessary by introduction of tree-dominated vegetation centered on the weir (Table 6 and Figure 3). 

The ecological condition of river reaches evaluated as “good” was greatly improved, as various woody plants were present. This decreases the ratio of areas occupied by exotic and obligate upland plants. Therefore, community and species diversities increased, and the vegetation profile improved greatly. However, non-ecological structures such as terraced floodplains continue to remain in the riverine structural frame; thus, fundamental improvement of the riverine structure is required for ecological restoration. In terms of vegetation, exotic and obligate upland plants continue to be found on the weir and should be especially replaced by native riparian vegetation (Table 6 and Figure 3). 

The ecological condition of river reaches evaluated as “very good” was relatively integrate, as the entire river range is covered with typical riparian vegetation with various plant species such as herbs, shrubs, and trees and complex stratification composed of tree, shrub, and herb layers. Thus, both community and species diversities are high, and the vertical structure expressed in the vegetation profile is equipped with diversity. Therefore, passive restoration, which is left to the natural process without any special treatment, is recommended here (Table 6 and Figure 3).

## 4. Discussion

### 4.1. Evolution of River Assessment Methods

The birth of the WFD began with a commitment to introduce a wide range of environmental policies common to the EU, including water resource protection. The European Commission (EC), which recognized the need to improve individual and fragmented water policies, proposed an integrated approach to water resources covering both the quantitative and qualitative aspects of the policy [62].

The WFD aims to protect and enhance aquatic ecosystems and promote sustainable water use across Europe. Water management is brought beyond water quantity and quality, entailing provisions on land-use and governance. The WFD sets environmental objectives in terms of good status to be met by 2015, or under certain conditions the final deadline of 2027 [62]. 

The WFD provides principles for assessing the status of entire aquatic ecosystems based on various biological elements or biological communities inhabiting waters [62,79]. The WFD sets forth provisions for the development of national assessment systems, with the novel requirement of taking into consideration reference conditions, according to which the degree of transformation in a given area of surface waters is assessed [62,79].

The WFD requires the assessment of different organism groups (i.e., benthic macroinvertebrates, benthic diatoms, aquatic macrophytes, and fish) called Biological Quality Elements (BQEs) to define the ecological status of rivers. These organisms were selected because they are widely considered good indicators of water quality, the alteration of which was the main pressure acting on rivers in developed countries in the last decades [62,80].

Each country evaluates the state of each country’s aquatic ecosystem according to the WFD standards, and then makes intercalations based on the regulations provided by Appendix V to ensure comparability among results from all over Europe. The foundation for intercalibration is formed by the normalized ecological quality ratios (EQRs), which represent the relationship between the values of the biological parameters observed in a given body of surface water and the values of these parameters in the reference conditions applicable to that body [62,79]. 

BQE-based metrics and indices that were developed for the implementation of the WFD and are used for standard assessment and monitoring are sensitive to water quality alteration and general habitat degradation. But their response to hydromorphological degradation is generally weak or absent [81,82,83,84]. Moreover, the effects of river restoration actions showed contrasting results on the BQEs richness and abundance [85]. The other significant shortcomings of the WFD-compliant biotic indices are also reported [86].

Most biotic and hydromorphological indices that have been developed for implementing the European Water Framework Directive (WFD) are characterized by limited spatial and temporal scales of application. The indices based on the Biological Quality Elements defined by the WFD are sensitive to water quality but not to hydromorphological alterations. Despite the significant monitoring effort, the effects of human pressures on ecosystems are poorly known. But the linkage between pressures and impacts is critical for developing sound measures that improve the ecological status in basins [87]. It is unreasonable to comprehensively assess the ecological conditions of the whole river corridor with exclusively BQE-based indices and metrics when we particularly consider the whole lateral dimension of the river system [88]. An incomplete evaluation could lead to incorrect planning in river management and restoration actions [86]. 

To overcome these limitations, alternative approaches, which can assess the horizontal dimension of river corridors and evaluate river corridor conditions comprehensively and find suitable management methods, are required [86].

The WFD has also acted as a driver for several interventions of river restoration aimed at the improvement of streams’ ecological conditions across the Member States [89]. The WFD asks to evaluate the chemical and the ecological status of their surface water bodies in a system by five scores ranging from high status to bad status. Water bodies with a status lower than “good” should achieve at least a good ecological status by improving ecological conditions. The benchmark for assessing river sections is the natural status, specified by river types. For water bodies which fail to reach the high or the good status (class I and class II) measures have to be implemented to achieve a good ecological status by 2015, with extension to 2021 and 2027 [62].

In this study, we evaluated the riparian vegetation in terms of a diagnostic evaluation for river restoration rather than monitoring for the health and integrity of rivers. Therefore, we assessed the river in a somewhat different perspective from the WFD system. We considered diversities of species, community, and functional groups as a health assessment factor for rivers, and evaluated the rivers by considering the proportion of the number and occupied area of exotic plant, obligate upland plant, and annual plant species as stress factors.

Riparian vegetation influences the ecological status of river ecosystems in many specific ways [18,42]. An improved understanding for the riparian vegetation will help in understanding and predicting ecological status of the river [90]. Moreover, one of the most important concerns in river restoration is how to deal with vegetation. The types and spatial arrangement of vegetation to be introduced are important in a river as an environment dominated by water currents. Moreover, vegetation is not only a biological component in a riverine environment but it also affects river morphology and functions in the habitat for other organisms. Thus, a diagnostic evaluation based on riparian vegetation is a significant preparatory step for river management that includes restoration [89,91]. 

### 4.2. Naturalness of the Rivers in Korea Based on Riparian Vegetation

Rivers, together with their marginal ecotonal systems, are corridors through the landscape; their margins provide buffers between a waterway and various land uses within a watershed. This affinitive relationship between land and water has been interrupted, degraded, and in extreme cases destroyed by human activity [71,92,93,94,95,96,97,98,99,100,101,102,103,104].

River systems have been dramatically altered by dams and reservoirs, channelization, and land-use development [71,93,96,97,98,99,100,101,102,103,104,105]. Some species of flora and fauna have disappeared and exotic species have invaded. The functional characteristics of river systems have been disrupted, and a reduction in landscape quality and loss of wilderness areas have been observed [71,96,97,98,99,100,101,102,103,104,105]. 

In Koreas, riparian landscapes are usually managed with a focus on use and disaster protection [71,96,97,98,99,100,101,102,103,104,106]. In Asian countries where people depend upon rice as a staple food, the floodplains of most rivers have been transformed into rice fields, and double terraced structure and high levees are constructed along waterways to prevent flooding. Consequently, the widths of most rivers have been sharply reduced, and terrestrialization has increased because of a decrease in the water table. Recently, many rice fields have been transformed into urban areas and naturally meandering and complex channels were forced into straight and monotonous lines. Because of such transformational processes, riverside communities have degenerated greatly or have been destroyed by tree cutting, introduction of exotic species, diversion and channelizing the watercourse for agriculture, and use of riverbeds and shores for cultivation or roads. The resulting riparian landscapes, including a river ecosystem and its surrounding environment, hardly retain any of their original features [54,55,96]. 

More than 70% of the evaluated reaches were assigned grades less than “moderate” (Figure 1). An evaluation based on each vegetation component—such as vegetation profile, community diversity, and species diversity—showed similar results (Figure 1). In the evaluation based on the exotic, obligate upland, and annual plants, grades based on the number of species ratio were poorer than the occupied area ratio. Therefore, these species invaded the riparian zone but did not expand so much as to form a community. However, if the current non-ecological riverine structure remains without any restorative treatments and artificial interferences continue to occur, those species would expand their territory and dominate the riparian zone, resulting in “very poor” or “poor” reaches. 

### 4.3. Necessity of River Restoration

Diagnostic assessment of a river based on riparian vegetation, which is not only a type of organism but also functions as an environment for other organisms, can provide significant basic information for river management [99,100,107]. The results of the diagnostic evaluation showed a higher number of poor reaches of lower naturalness grade than good reaches of higher grade (Figure 1).

This is due to low ecological diversity such as low community diversity and simple vegetation profile, and excessive invasion by obligate upland and exotic plants (Figure 1). Low ecological diversity is due to the reduced breadth of the river caused by the transformation of the floodplain into other land uses including rice paddy. In addition, reconstruction of the river cross section into the terraced type has also contributed to reducing the ecological diversity of the riparian zone. The double terraced cross section changes the gradient of the water table, thereby causing the invasion by the obligate upland and exotic plants [54,55,104]. 

This implies that the river has been degenerated to an overall weak and unstable ecological space. Therefore, ecological restoration for converting an artificial river to a natural river with natural features and functions is urgently required.

### 4.4. Recommendations for Achieving true Restoration

Riparian landscapes have been damaged by excessive use, resulting in a dramatic alteration of habitat structure and function. Historically, riparian landscapes have managed to promote human use and prevent disasters; however, recently, the importance of riverine environments as natural environments has gained increased attention [15,30,54,55,105,108,109,110,111,112]. 

Riparian ecosystems are composed of vegetation, habitats, ecosystems associated with water bodies, and they are dependent on the existence of perennial, ephemeral, or intermittent surfaces or subsurface drainage. Riparian ecosystems are some of the most productive ecosystems; they affect the stability and quality of the surrounding ecosystems indirectly by reducing flood peaks, acting as sediment and nutrient sinks, controlling water temperature, and increasing ground water recharge [31,33,35,36,113,114,115]. Therefore, despite their relatively small expanse, riparian areas play a critical role in the life cycles of an inordinate number of wildlife species and provide important recreation opportunities for outdoor enthusiasts [18,19,116,117].

Since the 1990s, river managers have used ecological methods to restore the communities and processes of river ecosystems in Korea [118]. However, such restoration projects usually focus on the waterfront without consideration to the entire range of the riverine landscape. Therefore, the ecological quality is very low, even in the river to which the restorative treatment has been applied [54,55,98,100,101]. 

Indeed, the restoration projects executed in Korea have many limitations. A series of procedures are required to achieve a successful ecological restoration. However, these procedures usually tend to be ignored in most restoration projects implemented in Korea. Diagnostic evaluation is generally omitted. Even if a diagnostic evaluation is made, there are very few cases in which the level and method of restorative treatment are determined based on the result, and most restoration projects are progressed only by active methods without any relation to the degree of damage [98,119]. Therefore, cost and energy are wasted and the effect is very little [98,104,120]. The results of this study provide diagnostic evaluation results of rivers attempted on a national scale (Figure 1). Thus, these results can serve as a means of prioritizing future restoration projects as well as information that determines the level and method of restoration (Table 6).

In most restoration projects, the reference information is not used and restoration is performed based on the subjective decisions of the project manager. Thus, restoration projects are conducted without any model or goals. Consequently, exotic species, which should be excluded thoroughly in a restoration project, are introduced frequently, and the spatial distribution range for plant species is barely considered [98,104,120]. This study also provides the reference ecological information required to perform ecological restoration of damaged rivers (Table 5). However, this information is generalized on a nationwide scale, but the results of this study can also function as local ecological reference information. For example, if the river to be restored is determined, the ecological information of the reach, which is evaluated as ‘very good’ grade among the reaches close to the river, could be used as the reference information (refer to Figure 1). Monitoring after restoration is performed; however, adaptive management is not performed because reference information as a restoration goal was not used in the preparatory stage of the project. In addition, evaluation of the restoration effects is not performed. Therefore, although restoration projects are continuously implemented, they have not evolved over the years [98,107,120].

Indeed, this non-systematic river restoration or management is common in developing countries such as Korea [98,119]. In this regard, the results of this study, which established a framework for determining the priority and level of restoration through systematic diagnostic evaluation and furthermore, prepared the reference ecological information for restoration to be implemented in the future based on vegetation data obtained from natural rivers, are meaningful.

Restoration efforts should be more effective; ecological restoration needs to be performed thoroughly, and the spatial range should be expanded so that floodplains, weirs, and surrounding environments, which are usually occupied by agricultural fields and/or urbanized areas, are included [54,55,121]. The biggest problem is that the spatial range of the river has been reduced due to excessive land use. Furthermore, even within the reduced space, the river is being transformed into double terraced structure for another use, and narrowing the width of the waterway again. In this cross-sectional structures, floodplain become drier, resulting in the flourishing of exotic species and obligate upland plants. Therefore, it is absolutely necessary to improve the cross-sectional structure in the river restoration to be progressed in the future, and furthermore, to secure the spatial range for equipping the integrate cross-section of the river [54,55]. It is necessary to imitate and accept the “re-profiling” project [122] and “room for the river” project [123] practiced already in Europe.

## 5. Conclusions

Asian countries, including Korea, where rice is a staple food, have lost many riparian ecosystems because of the transformation of most of the floodplains into rice paddies or urban areas. Consequently, the riverine landscape has been degraded to a poor and unstable state. The results of the diagnostic evaluation reflect this trend (Figure 1). The community and species diversities were low, and the vegetation profile was monotonous because herbaceous plants, including exotic or obligate upland plants, dominated the riparian vegetation, which is usually the result of a substantial reduction in the breadth of a river. Today, riparian ecosystems are reevaluated as very significant ecological spaces from various perspectives that include pollution control, biodiversity conservation, and disaster protection. Therefore, river restoration, including the entire range of the riparian landscape, is urgently required, and our diagnostic evaluation based on riparian vegetation could significantly contribute to the restoration and further maintenance of a healthy environment in a given area.

Restoration is an ecological technology that ameliorates degraded nature by imitating integrated and healthy nature. Restoration is achieved through a series of procedures, such as a survey of the existing conditions, statement of the goals and objectives, designation and description of a reference, preparation of a master plan, establishment of a restoration plan, restoration practices, monitoring, adaptive management, and evaluation [43,44,93,122]. In Korea, most restoration projects neglected such procedures and thus did not meet the restoration goals, in spite of great expense and labor [54,55,96,98,104,107,119,120]. This study provides results obtained from a diagnostic evaluation and a restoration plan based on the results. The diagnostic evaluation of the riparian vegetation reflects the necessity for ecological restoration in most river reaches in Korea. Furthermore, the results from the diagnostic evaluation can help to determine the priority and level of restoration. In addition, the reference information as another result of this study could serve as a model for planning ecological restoration projects for rivers and later help in the evaluation of the projects. 

## Figures and Tables

**Figure 1 ijerph-18-01724-f001:**
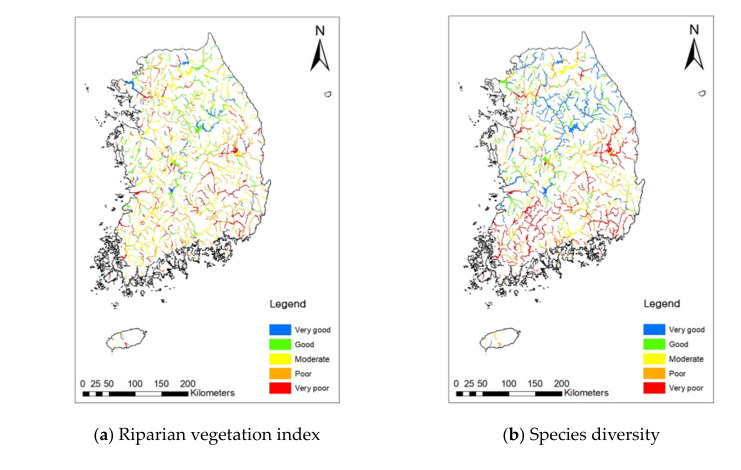
Nation-wide distribution of the grades based on each vegetation component and the riparian vegetation index of rivers in South Korea.

**Figure 2 ijerph-18-01724-f002:**
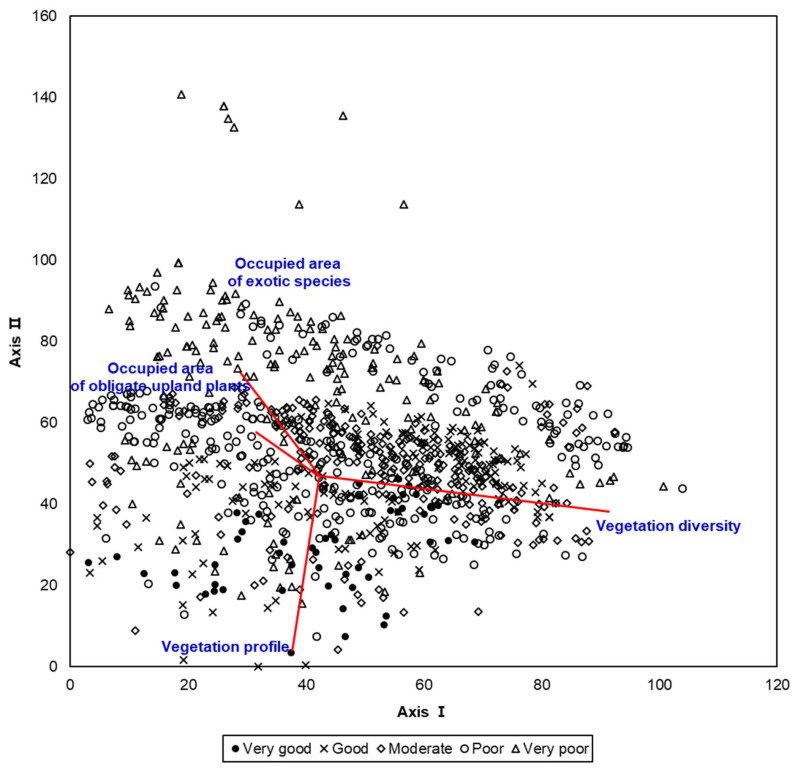
Stand ordination of 960 reaches, which were evaluated based on the riparian vegetation index.

**Figure 3 ijerph-18-01724-f003:**
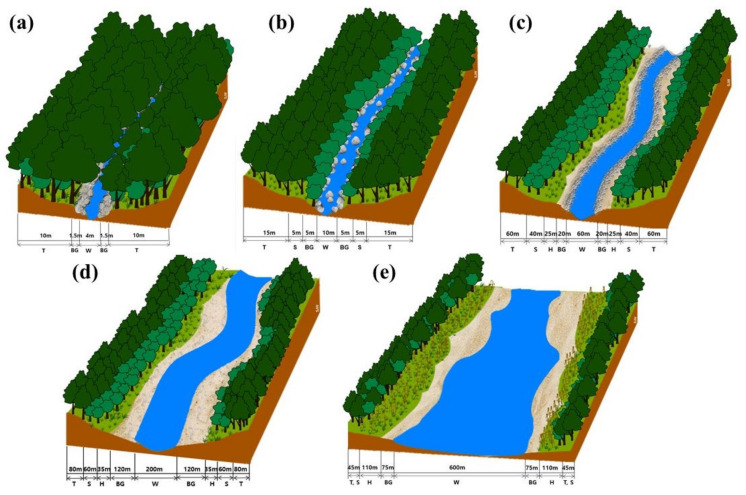
Spatial arrangement of riparian vegetation based on the reference information. Arrangement of landscape elements on the cross section of river appears in the order of bare ground, herb, shrub, and tree-dominated zones; zones reflecting flooding regime under the monsoon climate. Upper and lower valley streams, and downstream lack shrub and herb, herb, and shrub-dominated zones, respectively. This difference is determined by current speed due to the slope of riverbed. The breadth of each zone is a relative one and thereby it depends on the width of the waterway. Information on vegetation in each zone was suggested in Table 5. (**a**) upper valley stream, (**b**) lower valley stream, (**c**) upstream, (**d**) midstream, (**e**) downstream, W: waterway, BG: bare ground, H: herb dominated zone, S: shrub dominated zone, T: tree dominated zone.

**Figure 4 ijerph-18-01724-f004:**
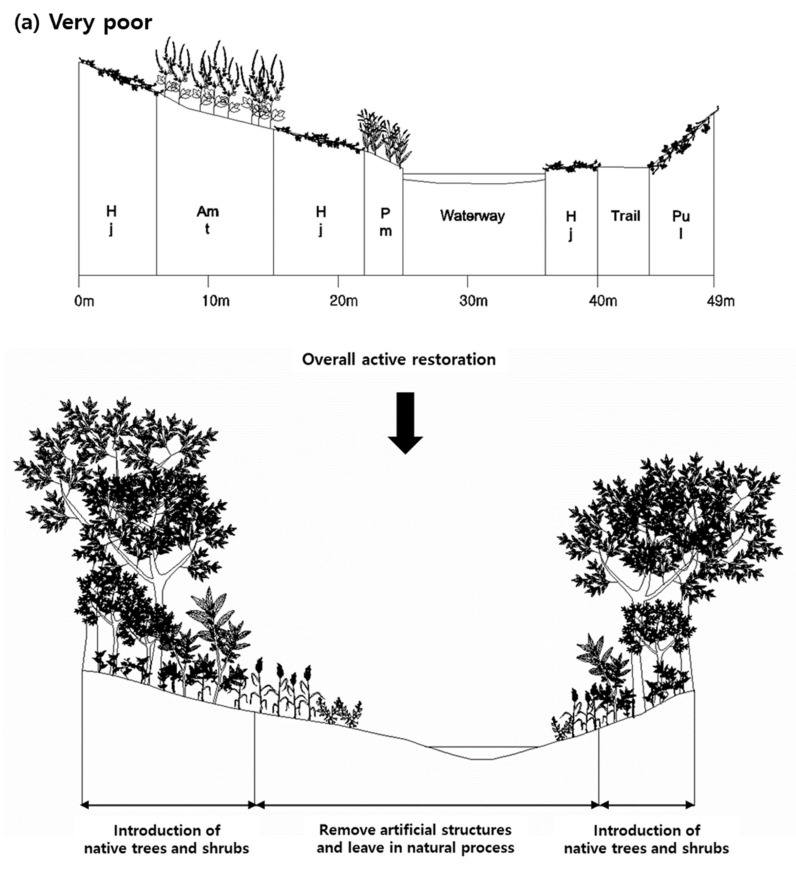
Ecological restoration plan prescribed based on the result of diagnostic evaluation. The lower the naturalness grade, the higher the level of restorative treatment was recommended. ‘Overall active restoration’, ‘partially active restoration’, ‘combination of active and passive restoration’, ‘artificial support’, and ‘passive restoration left in natural process’ were suggested for ‘Very poor’, ‘poor’, ‘moderate’, ‘good’, and ‘very good’ grades, respectively. It was recommended that introduction of riparian vegetation should follow reference information familiar with the reach and further, restrict on tree and shrub dominated zones and herb dominated zone where is exposed on natural disturbance such as flooding frequently, be left in natural process. Act: *Acer tataricum*, Ad: *Ajuga decumbens*, Amt: *Ambrosia trifida*, Ca: *Clematis apiifolia*, Coc: *Conyza canadensis*, Cs: *Carex scabrifolia*, Ec: *Echinochloa crusgalli* var. *oryzicola*, Fr: *Fraxinus rhynchophylla*, Hj: *Humulus japonicus*, Mb: *Morus bombycis*, Pa: *Phalaris arundinacea*, Pc: *Phragmites communis*, Pj: *Phragmites japonica*, Pm: *Persicaria modosa*, Pp: *Persicaria perfoliata*, Pt: *Persicaria thunbergii*, Pul: *Pueraria lobata*, Rj: *Rhus javanica*, Rp: *Robinia pseudoacacia*, Sg: *Salix gracilistyla*, Sk: *Salix koreensis*, Sv: *Setaria viridis*, Zl: *Zizania latifolia.* (**a**) Restoration plan of Anyang stream(H258) assessed as “very poor” in the naturalness grade. (**b**) Restoration plan of Gulji steam(H204) assessed as “poor” in the naturalness grade. (**c**) Restoration plan of Byeokgye stream(H215) assessed as “moderate” in the naturalness grade. (**d**) Restoration plan of Inbuk stream(H177) assessed as “good” in the naturalness grade. (**e**) A feature of Han River (H269) assessed as “very good” in the naturalness grade.

**Table 1 ijerph-18-01724-t001:** Components and scores applied to assess the naturalness of rivers based on riparian vegetation in South Korea. Numbers in parentheses indicate the weighted values.

Score Component	1	2	3	4	5
Community diversity (no. of communities, 4)	<3	4 to 5	6 to 7	8 to 9	>10
No. of exotic species (%, 1)	>34	25 to 34	15 to 24	5 to 14	<5
No. of annual plants (%, 1)	<9, >56	51 to 55	46 to 50	41 to 45	10 to 40
No. of obligate upland plant (%, 1)	>75	71 to 75	66 to 70	61 to 65	<61
Occupied area by exotic plants (%, 1)	>35	26 to 35	16 to 25	6 to 15	<6
Occupied area by annual plants (%, 1)	>49	37 to 49	24 to 36	11 to 23	<11
Occupied area by obligate upland plants (%, 1)	>71	55 to 71	38 to 54	21 to 37	<21
Vegetation profile (grade, 8) ^1^	First	Second	Third	Fourth	Fifth
Species diversity (no. of species, 2)	<31	31 to 40	41 to 50	51 to 60	>60

^1^ Grade of vegetation profile was based on [71].

**Table 2 ijerph-18-01724-t002:** Criteria for evaluating the naturalness of rivers based on score of riparian vegetation index in South Korea. Riparian vegetation index was obtained from the sum of score determined by multiplying the score and the weighted value of each vegetation component in Table 1.

Grade of Naturalness	Very Good	Good	Moderate	Poor	Very Poor
Score range	>70	61 to 70	51 to 60	41 to 50	<41

**Table 3 ijerph-18-01724-t003:** Correlations (Pearson) between ten vegetation components, including a synthetic index obtained from combining the naturalness and weighted values of each vegetation component.

	Species Diversity	Community Diversity	Vegetation Profile	No. of Exotic Plants	No. of Obligate Upland Plants	No. of Annual Plants	Occupied Area of Exotic Species	Occupied Area of Obligate Upland Plants	Occupied Area of Annual Plants
Community diversity	0.17 **								
Vegetation profile	0.35 **	0.11 **							
No. of exotic species	−0.03	−0.02	−0.08 *						
No. of obligate upland plants	0.00	−0.12 **	−0.10 **	0.08 *					
No. of annual plants	−0.07 *	0.09 **	−0.23 **	0.32 **	−0.29 **				
Occupied area of exotic species	0.09 **	0.16 **	0.02	0.29 **	0.09 **	0.03			
Occupied area of obligate upland plants	0.00	0.12 **	−0.14 **	021 **	0.14 **	0.08 *	0.44 **		
Occupied area of annual plants	−0.04	0.04	−0.23 **	0.17 **	−0.07 *	0.37 **	0.11 **	0.34 **	
Riparian vegetation index	0.52 **	0.51 **	0.76 **	−0.25 **	−0.16 **	−0.26 **	−0.11 **	−0.29 **	−0.35 **

*: *p* < 0.05; **: *p* < 0.01.

**Table 4 ijerph-18-01724-t004:** Correlations (Pearson) between riparian vegetation index and index based on other taxa and BOD. BMI: Benthic Macroinvertebrates Index, TDI: Trophic Diatom Index, FAI: Fish Assessment Index, HRI: Habitat Riparian Index, BOD: Biochemical Oxygen Demand.

Environmental Factors	BMI	TDI	FAI	HRI	BOD
RVI	0.18 **	0.26 **	0.19 **	0.16 **	−0.10 **

**: *p* < 0.01.

**Table 5 ijerph-18-01724-t005:** Species composition by horizontal zone of vegetation to be introduced for restoration. This reference information was prepared by incorporating riparian vegetation data collected from the river reaches evaluated as ‘very good’ grade in the diagnostic assessment for the naturalness of rivers executed throughout the whole national territory of South Korea.

Geographic Segment	Vegetation Zone	Vegetation Layer		
		Canopy and Understory Tree	Shrub	Herb
Upper Valley	Tree zone	*Acer pictum* subsp. *mono*, *Carpinus cordata*, *Cornus controversa*, *Fraxinus mandshurica*, *Fraxinus rhynchophylla*, *Juglans mandshurica*, *Magnolia sieboldii* etc.	*Celastrus flagellaris*, *Deutzia parviflora*, *Lindera obtusiloba*, *Staphylea bumalda*, *Stephanandra incisa* etc.	*Astilbe rubra*, *Carex lanceolata*, *Chrysosplenium grayanum*, *Corydalis speciosa*, *Dryopteris crassirhizoma*, *Festuca ovina*, *Geum japonicum*, *Impatiens noli-tangere*, *Lamium album* var. *barbatum*, *Ligularia fischeri*, *Thalictrum aquilegifolium* var. *sibiricum*, *Vicia amoena* etc.
Lower Valley	Tree zone	*Acer pictum* subsp. *mono*, *Acer tataricum* subsp. *ginnala*, *Carpinus cordata*, *Fraxinus mandshurica*, *Fraxinus rhynchophylla*, *Juglans mandshurica*, *Salix koreensis*, *Ulmus davidiana* var. *japonica* etc.	*Alangium platanifolium* var. *trilobum*, *Flueggea suffruticosa*, *Salix gracilistyla*, *Staphylea bumalda*, *Stephanandra incisa*, *Weigela subsessilis* etc.	*Boehmeria tricuspis*, *Corydalis speciosa*, *Dryopteris crassirhizoma*, *Impatiens textori*, *Persicaria longiseta*, *Persicaria thunbergii*, *Prunella vulgaris* var. *lilacina*, *Pteridium aquilinum* var. *latiusculum*, *Stellaria alsine* var. *undulata*, *Vicia unijuga* etc.
Upstream	Tree zone	*Acer tataricum* subsp. *ginnala*, *Fraxinus rhynchophylla*, *Juglans mandshurica*, *Salix chaenomeloides*, *Salix koreensis*, *Salix subfragilis*, *Ulmus davidiana* var. *japonica* etc.	*Salix gracilistyla*, *Salix integra* etc.	*Artemisia selengensis*, *Carex glabrescens*, *Miscanthus sacchariflorus*, *Oenanthe javanica*, *Persicaria hydropiper*, *Persicaria thunbergii*, *Phalaris arundinacea*, *Phragmites japonica*, *Scirpus radicans*, *Stellaria aquatica* etc.
	Shrub zone	-	*Rosa multiflora*, *Philadelphus schrenkii*, *Salix graciliglans*, *Salix gracilistyla*, *Salix integra*, *Staphylea bumalda* etc.	*Artemisia capillaris*, *Equisetum arvense*, *Miscanthus sacchariflorus*, *Oenanthe javanica*, *Persicaria thunbergii*, *Phalaris arundinacea*, *Phragmites communis*, *Phragmites japonica*, *Scirpus radicans* etc.
	Herb zone	-	-	*Artemisia selengensis*, *Barbarea orthoceras*, *Cardamine flexuosa*, *Carex neurocarpa*, *Corydalis speciosa*, *Impatiens noli-tangere*, *Impatiens textori*, *Miscanthus sacchariflorus*, *Oenanthe javanica*, *Persicaria nodosa*, *Persicaria thunbergii*, *Phalaris arundinacea*, *Phragmites japonica*, *Scirpus radicans*, *Viola verecunda* etc.
Midstream	Tree zone	*Morus alba*, *Salix chaenomeloides*, *Salix koreensis*, *Salix pseudolasiogyne*, *Salix subfragilis*, *Ulmus parvifolia*, *Salix chaenomeloides* etc.	*Salix gracilistyla*, *Salix integra*, *Salix koriyanagi*, *Rosa multiflora* etc.	*Artemisia selengensis*, *Carex glabrescens*, *Carex miyabei*, *Centella asiatica*, *Impatiens noli-tangere*, *Miscanthus sacchariflorus*, *Oenanthe javanica*, *Phalaris arundinacea*, *Phragmites communis*, *Phragmites japonica*, *Scirpus radicans* etc.
	Shrub zone	-	*Salix graciliglans*, *Salix gracilistyla*, *Salix integra* etc.	*Alopecurus aequalis*, *Carex glabrescens*, *Equisetum arvense*, *Miscanthus sacchariflorus*, *Persicaria thunbergii*, *Phalaris arundinacea*, *Phragmites communis*, *Phragmites japonica*, *Scirpus radicans*, *Stellaria aquatica*, *Typha orientalis* etc.
	Herb zone	-	-	*Artemisia selengensis*, *Centella asiatica*, *Echinochloa crusgalli* var. *oryzicola*, *Lespedeza cuneata*, *Miscanthus sacchariflorus*, *Penthorum chinense*, *Persicaria hydropiper*, *Persicaria nodosa*, *Phalaris arundinacea*, *Phalaris japonica*, *Phragmites communis*, *Phragmites japonica*, *Sium suave* etc.
Downstream	Tree zone	*Salix chaenomeloides*, *Salix koreensis*, *Salix pseudolasiogyne*, *Salix subfragilis* etc.	*Rosa multiflora*, *Salix gracilistyla*, *Salix koriyanagi*, *Salix integra* etc.	*Artemisia selengensis*, *Miscanthus sacchariflorus*, *Oenanthe javanica*, *Persicaria dissitiflora*, *Persicaria thunbergii*, *Phalaris arundinacea*, *Phragmites communis*, *Phragmites japonica*, *Scirpus radicans*, *Typha orientalis* etc.
	Herb zone	-	-	*Artemisia selengensis*, *Carex glabrescens*, *Carex scabrifolia*, *Cyperus amuricus*, *Echinochloa crusgalli* var. *echinata*, *Echinochloa utilis*, *Eclipta prostrata*, *Juncus effusus* var. *decipiens*, *Miscanthus sacchariflorus*, *Persicaria thunbergii*, *Phacelurus latifolius*, *Phalaris arundinacea*, *Phragmites communis*, *Phragmites japonica*, *Scirpus radicans*, *Viola verecunda*, *Zizania latifolia* etc.

**Table 6 ijerph-18-01724-t006:** Level and method of restoration recommended based on a diagnostic evaluation of the rivers throughout the whole national territory of South Korea.

Naturalness Degree	Degradation Degree	Restoration Level	Restoration Method
Very poor	Very severely degraded	Overall active restoration	Restoration throughout the entire range of the riparian landscape from the channel through the floodplain to the weir, including morphology of river, vegetation, watershed management, and networking with the surrounding terrestrial ecosystem.
Poor	Severely degraded	Partially active restoration	Similar restoration to the above but at a lower level Transformation of waterfront protection material from artificial to natural ones and introduction of vegetation on the floodplain and levee
Moderate	Moderately degraded	Combination of active and passive restoration	Transformation of waterfront protection material from artificial to natural ones and introduction of vegetation on some parts of the floodplain and the levee
Good	Lightly degraded	Artificial support	Transformation of waterfront protection material from artificial to natural ones and induction of establishment of natural vegetation on the levee
Very good	Natural	Passive restoration	Leave in its natural process

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
