# Peer review of "Diagnostic Evaluation and Preparation of the Reference Information for River Restoration in South Korea"

_ijerph, 2021, doi:10.3390/ijerph18041724_

Round 1

Reviewer 1 Report

I have read the revised resubmitted manuscript because the manuscript topic itself has merits but needed some improvements.

In resubmitted manuscript, Introduction and Discussion have been improved.

But resubmitted manuscript's research aims, results and discussion are not well matching. So Please keep consistency of topics throughout the whole manuscript.

And don't jump at the discussion and avoid redundancy in the manuscript.

Just describe what authors have found from the your research data and well define the terms you have used in the manuscript.

Summary, your resubmitted manuscript has been improved from the previous one, but still needs keeping consistency of topics and terms throughout the Introduction, Results and Discussion sections. 

Author Response

Thank you for your considerate comments. We revised our manuscript by accepting your comments.

Thank you.

Best regards.

Reviewer 2 Report

I think the manuscript has been improved and now warrants publication in IJERPH.

Author Response

Thank you for your considerate comments.

Best regadsa.

Reviewer 3 Report

Dear authors, I made comments to the editor who will provide you with a final comment

I recommend to carefully analyze the points I marked in the previous revision, one by one, because there still are several pending issues.

kind regards

Author Response

Thank your for your considerate comments.

We revised our manuscript by accepting your comments.

Best regards.

This manuscript is a resubmission of an earlier submission. The following is a list of the peer review reports and author responses from that submission.

Round 1

Reviewer 1 Report

This manuscript is dealing with diagnostic evaluation and systematization of the reference information for river restoration. The methods are sound and results are well represented. A few suggestions for the better scientific paper.

  1. Clarify the term of vegetation types (no. of types)
  2. In methods, no. of obligate upland plant >61 all belong to category 5. What % of plots are belong to category 5? And category 1?
  3. In the manuscript the authors are using river or rivers. Does he term river include river and stream? Please define the term of river in you manuscript.
  4. The most important question. Is this diagnostic evaluation system can be applied for other regions of world? Or only can be applied for your country?

Author Response

Response to reviewer’s comments

Reviewer 1 

 Comments and Suggestions for Authors

This manuscript is dealing with diagnostic evaluation and systematization of the reference information for river restoration. The methods are sound and results are well represented. A few suggestions for the better scientific paper.

  1. Clarify the term of vegetation types (no. of types)

Vegetation type means plant community. So, the number of types indicates the number of plant communities surveyed in the selected river reach.

  1. In methods, no. of obligate upland plant >61 all belong to category 5. What % of plots are belong to category 5? And category 1?

The number of 61 means that the number of obligate upland plant species occupied 61% among the total number of plant species surveyed in the selected river reach. We allocated the reach that the percentage less than 61% to category 5 and the reach that the percentage less than 75% to category 1.

  1. In the manuscript the authors are using river or rivers. Does he term river include river and stream? Please define the term of river in you manuscript.

We usually classify river, stream, and creek according the scale but we didn’t classify them. But when we the river reaches depending on the geographic position, we used the term of stream such as valley stream, upstream, midstream, and downstream.

  1. The most important question. Is this diagnostic evaluation system can be applied for other regions of world? Or only can be applied for your country?

Of course, this diagnostic evaluation system can be applied worldwide. As you know, our diagnostic evaluation system is focused on diversity such as species, community, and functional diversities, which imply the stability. Then, we considered disturbance factor, which suggests the instability such as the number of species and the occupied area of exotic, obligate upland, and annual plants.

Reviewer 2 Report

The naturalness of rivers in entire South Korea were assessed based on the riparian vegetation index and vegetation elements, based on the results of assessment, the restoration plans were also proposed. The results including many good quality figures for the reference to restore the degraded river ecosystem were interesting and useful for South Korea and other similar area in the world, especially in Asia. I recommended it publish after minor revision.

1. Although the reason of riparian vegetation degradation was mentioned (human disturbance), it is better to explain in the section of “Discussion” what part of river or what kind of river environment are easy to be disturbed or damaged by human disturbance, therefore, it can help local decision makers to reduce the disturbance in such areas.

2. The list of reference was to long, please reduce it.

Author Response

Response to reviewer’s comments

Reviewer 2

Comments and Suggestions for Authors

The naturalness of rivers in entire South Korea were assessed based on the riparian vegetation index and vegetation elements, based on the results of assessment, the restoration plans were also proposed. The results including many good quality figures for the reference to restore the degraded river ecosystem were interesting and useful for South Korea and other similar area in the world, especially in Asia. I recommended it publish after minor revision.

  1. Although the reason of riparian vegetation degradation was mentioned (human disturbance), it is better to explain in the section of “Discussion” what part of river or what kind of river environment are easy to be disturbed or damaged by human disturbance, therefore, it can help local decision makers to reduce the disturbance in such areas.

We revised our manuscript by accepting the reviewer’s comment. Lines 452-460.

  1. The list of reference was to long, please reduce it.

  The references are the necessary literature to explain our research. Therefore, if there is no special problem, I want to maintain the current state.

Reviewer 3 Report

Dear authors

I think your work deserves attention and merits a publication because it presents a nation-wide effort which definitely is a valuable experience.

However, I had several difficulties in understanding your paper and it rose so many doubts and questions recalling me the years of heavy discussions within the Italian Center for River Restoration which I directed for almost 10 years...

I am sorry hence to convey to the Editor the message that the paper , in my opinion, has to be improved. But I sincerely hope to see it in a more robust and straightforward-to-read version, suited to be published.

Here below go my main comments. I also attach the paper with several minor comments, hoping to help you further. Kind regards

- The title speaks of Reference information, but from the Abstract a restoration plan issue is introduced which plays even a major role within the paper. Actually, from the title, I thought the central issue would be the problem of how to establish Reference Status. I see hence the need to modify the title to capture this other issue
- From the Abstract and later on the term "vegetation elements" is used, but I could not understand what is it and its role

- Introduction: this section should first go through other experiences and only
afterward state what you did and why. For example, lines 96-101 interrupt
the presentation of the work done. Moreover, it is not clear what is the
added value as compared to existing experiences: a new method? covering a nation with an assessment still missing (I guess so, stressing that it is a national level study)? Unexpected findings? Valuable indications for management? Suggestions for a similar exercise elsewhere?...

- The Naturalness criterion adopted is questionable or at least needs some further explanation and discussion. In Europe, in particular, the Water Framework Directive introduced a clear (although not easy to apply) criterion which is measuring the closeness to a Reference Status (RS), meaning with RS the conditions of a reach of a similar typology, but untouched by human alterations. So the first problem became establishing such RS for all different river typologies and additionally ensure a uniform meaning across all countries in Europe ("intercalibration").
- Another important issue is that considering just the vegetational status to assess river ecological status (which I understand you make coincide with "naturalness") and, on this basis, establishing a restoration strategy, is in my opinion a too partial point of view. I would expect to see several other key aspects considered like river hydromorphology (lateral space, longitudinal continuity, flows alteration...), water quality, biota; artificialization (hydraulic works, dams,...). Later in the paper, it is said that a correlation has been analyzed between the vegetation indicators adopted and other aspects like water quality (BOD only?), macroinvertebrates, ...But that does not satisfy this issue
- Concerning the Methodology, at lines 166-167 it seems that you first segmented the river according, basically, to its slope (which would be a quite rough method to determine meaningful river segments) and then established the RS by adopting the best scoring reach in each segment and then you measured a kind of distance from your vegetation index and such RS. It is important to ensure whether this is the actual method and explain it better (I may have given a wrong interpretation)
- Par 3.4 Systematization seems again an explanation of how you defined RS for different typologies, but it is not clear and it does not seem to cover all
aspects, while if you are defining RS ...that should include all attributes. And all this should appear in one place only where you first determine river segments and then determine their RS (if this is the case)
- Technically, I could not understand how you pass from indicators (Table 1) to an index of which you just give a classification (Table 2). There is a short verbal description, but it is not clear and certainly, it would be fair to formalize this step
- Table 6: it is not clear the difference between the first two fields. And I would have expected a differentiation in terms of scales of action: local, reach, river corridor, river basin: the first (worst) case mentions indeed river basin, but it is the only case and seems quite generic. Additionally, Fig.4 is a bit weird because i) it includes the restoration of cases in "Good" conditions, where I would expect preservation or conservation only; and ii) although based on the vegetation conditions, sketches show a significant change in topographic profile, not commented.
- The Discussion section is quite generic and could be absorbed by the Introduction. Actually, there is very little or no discussion about the work carried out: novelty, differences with other approaches-experiences; expectations and difficulties in its implementation; and nothing about monitoring strategy.

Author Response

Response to reviewer’s comments

Reviewer 3 

Comments and Suggestions for Authors

Dear authors

I think your work deserves attention and merits a publication because it presents a nation-wide effort which definitely is a valuable experience.

However, I had several difficulties in understanding your paper and it rose so many doubts and questions recalling me the years of heavy discussions within the Italian Center for River Restoration which I directed for almost 10 years...

I am sorry hence to convey to the Editor the message that the paper , in my opinion, has to be improved. But I sincerely hope to see it in a more robust and straightforward-to-read version, suited to be published.

Here below go my main comments. I also attach the paper with several minor comments, hoping to help you further. Kind regards

- The title speaks of Reference information, but from the Abstract a restoration plan issue is introduced which plays even a major role within the paper. Actually, from the title, I thought the central issue would be the problem of how to establish Reference Status. I see hence the need to modify the title to capture this other issue

We revised our manuscript by accepting the reviewer’s comment. Lines 2-46.,

- From the Abstract and later on the term "vegetation elements" is used, but I could not understand what is it and its role

We revised our manuscript to reduce misunderstanding of the readers as the follows: The naturalness in rivers was evaluated from the perspectives of both each vegetation element including species diversity, community diversity, vegetation profile, the number and the occupied area of exotic, obligate upland, and annual plant species, and riparian vegetation index obtained by synthesizing the elements. Lines 141-144.

- Introduction: this section should first go through other experiences and only
afterward state what you did and why. For example, lines 96-101 interrupt
the presentation of the work done. Moreover, it is not clear what is the
added value as compared to existing experiences: a new method? covering a nation with an assessment still missing (I guess so, stressing that it is a national level study)? Unexpected findings? Valuable indications for management? Suggestions for a similar exercise elsewhere?...

We explained the methods that other researchers used when they evaluate the river and the riparian conditions in lines 88 to 91. Then, we explained the background that we selected our evaluation method in lines 92 to 95.

- The Naturalness criterion adopted is questionable or at least needs some further explanation and discussion. In Europe, in particular, the Water Framework Directive introduced a clear (although not easy to apply) criterion which is measuring the closeness to a Reference Status (RS), meaning with RS the conditions of a reach of a similar typology, but untouched by human alterations. So the first problem became establishing such RS for all different river typologies and additionally ensure a uniform meaning across all countries in Europe ("intercalibration").

In Asian countries where rice, an aquatic plant, is the staple food, the intensity of land use around rivers is much stronger than that in European countries where wheat is the staple. Therefore, the width of the stream has been greatly reduced, and it is very difficult to find a river with integrate structure. Therefore, we currently conduct a nationwide survey like this and take the section with the highest grade as the reference state and evaluate and manage rivers based on it in the future. We carried out this study for that purpose.
- Another important issue is that considering just the vegetational status to assess river ecological status (which I understand you make coincide with "naturalness") and, on this basis, establishing a restoration strategy, is in my opinion a too partial point of view. I would expect to see several other key aspects considered like river hydromorphology (lateral space, longitudinal continuity, flows alteration...), water quality, biota; artificialization (hydraulic works, dams,...). Later in the paper, it is said that a correlation has been analyzed between the vegetation indicators adopted and other aspects like water quality (BOD only?), macroinvertebrates, ...But that does not satisfy this issue

You are right. My colleagues are carrying out studies for those factors. They may submit the results to this special issue. Our team focused on the riparian vegetation. Afterward, we will collect those data and review them comprehensively.
- Concerning the Methodology, at lines 166-167 it seems that you first segmented the river according, basically, to its slope (which would be a quite rough method to determine meaningful river segments) and then established the RS by adopting the best scoring reach in each segment and then you measured a kind of distance from your vegetation index and such RS. It is important to ensure whether this is the actual method and explain it better (I may have given a wrong interpretation)

As I mentioned above, it is very difficult to find a river with integrate structure due to excessive land use around rivers in Asian countries where rice is the staple food. Based on previous studies on the ecological characteristics of rivers, it was usually a tendency to be divided into five sections. The dominant factor in such distinctions was gradient of the river bed, which is the ratio of the elevation changing along the moving distance. In each divided section, we took the section with the highest naturalness as the reference state. Based on the results of the naturalness evaluation, we recommended active restoration for the section evaluated as the low naturalness degree and to left it in natural process or a low-level restoration for the section of high naturalness. Lines 179 – 187.

- Par 3.4 Systematization seems again an explanation of how you defined RS for different typologies, but it is not clear and it does not seem to cover all
aspects, while if you are defining RS ...that should include all attributes. And all this should appear in one place only where you first determine river segments and then determine their RS (if this is the case)

We classified the river into five reaches of upper and lower valley streams, upstream, midstream, and downstream based on substrate as well as the riverbed gradient obtained from the relationship between distance from the river mouth and elevation above sea level. Then, we took the section with the highest naturalness in each divided section as the reference reach. We synthesized vegetation data collected in the reference reach and we took the data as the reference information. The information includes the spatial range, dominant species and species composition of vegetation zones such as grassland, shrubby forest, and tree forest determined depending on the flooding regime.

Breadth of each zone is relative one and thereby it depends on the width of the waterway. Lines 293 – 315.
- Technically, I could not understand how you pass from indicators (Table 1) to an index of which you just give a classification (Table 2). There is a short verbal description, but it is not clear and certainly, it would be fair to formalize this step

We explained the process in detail from line 141 to 168.

- Table 6: it is not clear the difference between the first two fields. And I would have expected a differentiation in terms of scales of action: local, reach, river corridor, river basin: the first (worst) case mentions indeed river basin, but it is the only case and seems quite generic.

Such details were expressed by reducing and generalizing the contents, judging that they were suitable for the design for the restoration but were not suitable for the research paper.

Additionally, Fig.4 is a bit weird because i) it includes the restoration of cases in "Good" conditions, where I would expect preservation or conservation only; and ii) although based on the vegetation conditions, sketches show a significant change in topographic profile, not commented.

As I mentioned above, land use intensity around rivers in Asian countries where rice is the staple food is very high. So, we recommended leaving them in natural process, that is passive restoration just for the reaches, which are assessed as “very good” grade and the restorative treatment depending on the degradation degree for the reaches of grade lower than that.

and ii) although based on the vegetation conditions, sketches show a significant change in topographic profile, not commented.

It’s our mistake. We revised it.

- The Discussion section is quite generic and could be absorbed by the Introduction. Actually, there is very little or no discussion about the work carried out: novelty, differences with other approaches-experiences; expectations and difficulties in its implementation; and nothing about monitoring strategy.

We revised ‘Discussion’ section by accepting reviewer’s comment. Lines

Minor comments on Manuscript

with no doubts riparian vegetation supports a number of services as yu mention; but I would not hide also undersired effects. For instance, by increasing roughness, vegetation can increase water levels and so local flooding (it may protect downstream, but worsen locally or upstream). Also, big trees may fall down and deviate the water course with tremendous impacts. Wood, branches may accumulate and block the flow through (insufficient) bridges...

☞ As we mentioned in our manuscript (Lines 376 – 386, 465 – 468), Asian countries, including Korea, where rice is a staple food, have lost many riparian ecosystems because of the transformation of most of the floodplains into rice paddies or urban areas. Consequently, the riverine landscape has been degraded to a poor state. Therefore, there are no riparian vegetation that would have an undesirable effect. For this reason, we did not mention those undesirable effects.

It would be nice to insert a figure showing this arrangement

It is a very general content and it is judged that no separate figure is needed.

what do you mean by weirs? Perhaps levees?

We revised it by accepting reviewer’s comment. Line 132.

from this information one cannot understand the density and coverage of field sampling: it is very important to explain that, possibly including a map and measured indicators

They are general method for vegetation survey and vegetation mapping.

both sentences are unclear: “synthesizing the elements…what are the elements? “based on the vegetation”?

We revised our manuscript to reduce misunderstanding of the readers as the follows: The naturalness in rivers was evaluated from the perspectives of both each vegetation element including species diversity, community diversity, vegetation profile, the number and the occupied area of exotic, obligate upland, and annual plant species, and riparian vegetation index obtained by synthesizing the elements. Lines 141-144.

Also, possibly because of my ignorance, but I do not understand what are “annual plants” (you mean with a one-year life cycle?) and “obligate upland plants” … cannot figure out what that is

Yes, you are right. An annual plant is a plant that completes its life cycle, from germination to the production of seeds, within one growing season, and then dies. Obligate upland plant almost always occurs in uplands under natural conditions (estimated probability > 99%).

This whole paragraph has been repeated at least three times including the Abstract; but still I cannot understand exactly what has been done

We revised our manuscript to aid understanding of the readers. Lines 160 – 169. We add a detailed description below.

We first divided the measured values for each vegetation element such as species diversity, community diversity, vegetation profile, the number and the occupied area of exotic, obligate upland, and annual plant species into five equal intervals, giving each a score between 1 and 5. Then, we gave weighted values for each element. A weighted value of 1 point was given to the percentage based on the number of species and occupied areas of exotic, annual, and obligate upland plants. We assigned weighted values of 2 and 4 points for species diversity, which addresses the composite factor related to various species, and community diversity, which addresses the composite factor related to various vegetation types as a two-dimensional element, respectively. The vegetation profile expresses the horizontal and vertical diversity of vegetation; 8 points were conferred to this element. The riparian vegetation index is obtained from the sum of the scores multiplied by the weighted value of each vegetation element. The riparian vegetation index was divided into five grades of “very good,” “good,” “fair,” “poor,” and “very poor”.

The title seems not appropriate

We revised it by accepting reviewer’s comment. Line 178.

habitat conditions mentioned above are not “taxa” either…

We revised our manuscript by adding them to aid understanding of the readers. Lines 193 – 194.

definition is needed

We revised it by accepting reviewer’s comment. Lines 276 - 277.

This part seems to describe part of the method, including the figure; as such it should have been presented in the previous section, not here in Results

This content is synthesizing the results obtained through the study. Therefore, it is judged to be a result rather than a method, so we maintained the current state.

I guess here there is a implicit order from top-left to bottom-right corresponding to mountain until lowlands … but each figure should be labelled and explained

We revised it by accepting reviewer’s comment. Figure 3.

This is the first time this (important?) issue is touched and it does not appear in Fig. 4 at all

We revised it by accepting reviewer’s comment. Figure 4.

Round 2

Reviewer 3 Report

please see Editors point of view